# Inflammation in Parkinson’s Disease: Mechanisms and Therapeutic Implications

**DOI:** 10.3390/cells9071687

**Published:** 2020-07-14

**Authors:** Marta Pajares, Ana I. Rojo, Gina Manda, Lisardo Boscá, Antonio Cuadrado

**Affiliations:** 1Instituto de Investigaciones Biomédicas “Alberto Sols” UAM-CSIC, 28029 Madrid, Spain; mpajares@iib.uam.es (M.P.); airojo@iib.uam.es (A.I.R.); lbosca@iib.uam.es (L.B.); 2Centro de Investigación Biomédica en Red Sobre Enfermedades Neurodegenerativas (CIBERNED), ISCIII, 28031 Madrid, Spain; 3Instituto de Investigación Sanitaria La Paz (IdiPaz), 28029 Madrid, Spain; 4Department of Biochemistry, Faculty of Medicine, Autonomous University of Madrid, 28029 Madrid, Spain; 5Department Cellular and Molecular Medicine, Victor Babes National Institute of Pathology, 050096 Bucharest, Romania; gina.manda@gmail.com; 6Centro de Investigación Biomédica en Red Sobre Enfermedades Cardiovasculares (CIBERcv), ISCIII, 28029 Madrid, Spain

**Keywords:** Parkinson’s disease, neuroinflammation, immune system, therapy, neurodegeneration

## Abstract

Parkinson’s disease (PD) is a common neurodegenerative disorder primarily characterized by the death of dopaminergic neurons that project from the substantia nigra *pars compacta*. Although the molecular bases for PD development are still little defined, extensive evidence from human samples and animal models support the involvement of inflammation in onset or progression. However, the exact trigger for this response remains unclear. Here, we provide a systematic review of the cellular mediators, i.e., microglia, astroglia and endothelial cells. We also discuss the genetic and transcriptional control of inflammation in PD and the immunomodulatory role of dopamine and reactive oxygen species. Finally, we summarize the preclinical and clinical approaches targeting neuroinflammation in PD.

## 1. Introduction

Parkinson’s disease (PD) is a common neurodegenerative disorder primarily characterized by the deterioration of motor activities due to the impairment of the dopaminergic nigrostriatal system. Specifically, the death of dopaminergic neurons that project from the substantia nigra *pars compacta* to the caudate-putamen in the striatum results in the loss of dopamine neurotransmission, causing the primary motor symptoms, including tremor at rest, bradykinesia, rigidity and postural instability. Although PD was initially described as a movement disorder without dementia, it is now accepted that PD progression affects other extra-nigral dopaminergic, cholinergic and serotoninergic tracts, leading to nonmotor symptoms that include anosmia, sleep disorders and constipation as well as cognitive and psychiatric symptoms, such as dementia and depression [1,2].

The onset of the disease occurs decades before the first symptoms appear; however, the ultimate underlying cause(s) of dopaminergic death remain unknown. While 5–10% of PD cases are of genetic origin (mutations in the *PARK* genes encoding alpha-Synuclein, DJ-1, PINK, LRRK2, etc.), leading to early PD onset, most cases remain idiopathic and are associated with aging. In addition to genetic predisposition, there are risk factors associated with environmental toxins, pesticides, heavy metals, traumatic lesions and bacterial or viral infections [3], all of which are closely associated with inflammation. A crucial role of inflammation in developing Parkinsonian symptoms has been suspected for many years since the initial observation of Parkinson-like symptoms in individuals infected with the influenza virus (encephalitis lethargica) [4]. Thereafter, PD onset has been associated with several viral pathogens including influenza A, Herpes simplex virus-1 (HSV-1), Ebola virus (EBV), varicella-zoster virus, Japanese encephalitis virus, vanilla necrosis virus, human immunodeficiency virus as well as with *Helicobacter pilori* [5]. Neurotropic pathogens might reach the basal ganglia through the nasal mucosa, via the olfactory pathways, and through the intestinal mucosa, via enteric plexuses and preganglionic vagal fibers, ultimately leading to a cascade of neuroinflammatory and neurodegenerative events in the nigrostriatal tract [6]. It is interesting that, in addition to eliciting an intense immune response in the nigrostriatal tract, some viral proteins (HSV-1 and EBV) exhibit molecular mimicry with alpha-Synuclein (α-Syn) and promote α-Syn aggregation and further accumulation in intracellular deposits called Lewy bodies [7]. Moreover, in the enteric nervous system α-Syn exhibits potent chemoattractant activity of neutrophils and monocytes in response to viral infection, further implying this protein in systemic and brain inflammatory responses in the pathogenesis of PD [8].

Inflammation is a highly regulated mechanism against pathogenic stimuli or tissue injury, aimed at protecting the host from the damage-causing agents and to promote tissue repair. The central nervous system (CNS) has long been considered an immune-privileged tissue due to the separation from the peripheral immune system by the blood–brain barrier (BBB). However, this concept has been revised. In fact, a potent innate immune response against Pathogen-Associated Molecular Patterns (PAMPs) and Damage-Associated Molecular Patterns (DAMPs) can be elicited in the CNS. Under physiological conditions, microglia and astroglia, constantly surveil the brain parenchyma to maintain CNS homeostasis by releasing neurotrophic factors, removing synaptic glutamate, remodeling and reshaping synapses, etc. However, these glial cells can be activated by PAMPs and DAMPs, such as secreted factors from damaged neurons or protein aggregates, leading to persistent neuroinflammation. Although it may not represent an initiating factor in all-case PD, chronic neuroinflammation appears to be a cofactor for disease progression [9].

As further discussed, extensive evidence in human samples and from animal models support the involvement of inflammation in the development of PD. However, the exact trigger for this response remains unclear. While inflammation might be the consequence of ongoing neuronal cell death in PD, it is also likely that misfolded α-Syn might play a direct role. In addition to the well-documented microgliosis and astrogliosis in PD brains, peripheral inflammation and PD-risk-associated genes support an important contribution of the chronic inflammatory response on the progression of this neurodegenerative disease. We will provide a systematic review of the cellular mediators and molecular mechanisms involved in neuroinflammation and how they may impact PD progression.

## 2. Role of Microglia

Microglia are CNS-resident macrophages, initially described by Pio del Rio Ortega [10]. These cells, in addition to homeostatic functions, constitute the frontline defense of the innate immune system. These cells secrete neurotrophic factors, remove toxic substances, and participate in neuronal repair, remodeling, and synaptic pruning [11]. The analysis of postmortem samples and neuroimaging implicate the microglia in PD pathophysiology [12].

Microglial dynamics along with several functional phenotypes allows a rapid and customized response to inflammatory stimuli. Under basal homeostatic conditions, microglia display a basal state of activity, inaccurately called resting state, in which they survey the microenvironment for sensing infection and cellular distress and respond to immunomodulatory molecules, such as CX3CL1, CD200, CD22, CD45, CD95 or NCAM that are released by healthy neurons [13]. In response to inflammatory challenges, microglia get functionally polarized towards the classical proinflammatory M1 phenotype or the alternative immunosuppressive and cytoprotective (“wound healing”) M2 phenotype [14]. Transcriptomic studies have added more levels of complexity to disease-specific microglial signatures, but these results still need to be validated [15]. DAMPs released by dying neurons (ATP, neuromelanin, m-calpain, matrix metalloprotease 3), or proinflammatory mediators secreted by astrocytes (CCL2), misfolded or aggregated proteins, such as α-Syn, along with signals delivered by Toll-like receptors (TLRs) favor the acquisition of the M1 proinflammatory phenotype by microglia [16,17,18]. This phenotype is characterized by a large cell body and amoeboid morphology as well as by upregulation of major histocompatibility complex (MHC) I and II molecules [19], and increased production of proinflammatory mediators (cytokines, i.e., IL1β, IL6, TNFα; chemokines and bioactive lipids) [20]. Therefore, M1 microglia can alter the BBB permeability and induce brain infiltration by circulating leukocytes [21], hence reinforcing the local inflammatory response. Additionally, DAMPs released by stressed and dying neurons induce the expression of genes that encode components of the NADPH oxidase (NOX) system and generate reactive oxygen species (ROS) and nitric oxide (via NOS1 and NOS3, rather than NOS2, in humans [22]), with antimicrobial function, that may sustain concurrently chronic inflammation [23]. Negative feedback mechanisms involving anti-inflammatory factors (cytokines such as IL4 and IL10, and the new class of proresolving mediators, comprising lipoxins, resolvins, protectins and maresins) are supposed to act for the resolution of inflammation under physiologic conditions, but might be deficient in chronic pathologies [24]. In agreement with the acquisition of the M1 microglial phenotype in PD, elevated levels of IL1β, IL6 and TNFα have been detected in the striatum, as well as in the SN of postmortem samples [25]. Interestingly, incubation of dopaminergic neurons with conditioned medium from M1 microglia was shown to favor cell death, whereas a mixture of medium from M1 and M2 microglia partially reversed neurotoxicity [26]. In another study, conditioned medium from LPS-treated glial cells containing IL1β and TNFα induced the death of cultured dopaminergic rat neurons [27]. Moreover, it has been shown that exosome release is enhanced in midbrain slices by microglial stimulation with LPS/IFNγ [28], and microglia-derived exosomes transfer α-Syn to neurons and increase neuronal apoptosis [29].

Preclinical studies in different animal models of PD reproduce these observations and have been useful to better understand the process of neuroinflammation in human PD. One of the most used animal models of PD consists of the striatal injection of 6-hydroxydopamine (6-OHDA), a catecholaminergic neurotoxin that is taken by dopaminergic neurons through the dopamine transporter and causes degeneration of the nigrostriatal system [30]. Injection of 6-OHDA in rats generates a reactive microgliosis that precedes the onset of astrogliosis and dopaminergic cell death. Different studies support the upregulation of proinflammatory mediators (i.e., IL1β, IL6, TNFα, IFNɣ) and its receptors, together with a decrease in anti-inflammatory molecules (IL10) [31]. The compound 1-methyl-4-phenyl-1,2,3,6-tetrahydropyridine (MPTP), a byproduct of synthetic heroine, is also taken by dopaminergic neurons and induces Parkinsonism in humans, nonhuman primates and mice. MPTP is converted to MPP^+^ by astrocytes. MPP^+^ is taken by dopamine receptors in neurons, where it inhibits the mitochondrial complex 1 of the electron transport chain, leading to ATP depletion and oxidative stress. These events finally result in dopaminergic cell death and activation of proinflammatory microglia [30]. In fact, both humans and nonhuman primates exposed to MPTP, exhibited active M1 microglia several years after MPTP exposure, suggesting a long-lasting and self-driven reactive microgliosis [32].

The rodent models based on 6-OHDA or MPTP intoxication reproduce the death of dopaminergic neurons but fail to show the main hallmark of most PD cases, which is the protein aggregation in Lewy bodies. Therefore, additional evidence of exacerbated microgliosis and proinflammatory cytokine levels has also been gathered from genetically modified animals expressing human wild type or mutant α-Syn [33,34,35]. Delivery of α-Syn to the nigrostriatal tract has been achieved either by midbrain infection with α-Syn-expressing adeno-associated viral vectors [36,37] or, more recently, by direct intrastriatal inoculation of preformed fibrils of α-Syn (PFF-α-Syn) which have the capacity to induce aggregation on endogenous α-Syn into Lewy-like aggregates [38]. Indeed, injection of PFF-α-Syn in the SN led to microglial activation and increased MHC class II molecules both in microglial cells and in newly recruited blood monocytes and lymphocytes [39]. Similar studies have shown increased IL1β, TNFα and IFNɣ in the striatum of α-Syn inoculated mice [40] and altered central and peripheral immune profiles [41]. The phagocytic capacity of microglia plays a critical role in α-Syn metabolism and, therefore, in PD development [42]. Studies performed in the microglial cell line BV2 suggested that extracellular α-Syn is detected and internalized via GM1 gangliosides as well as by an unknown protein receptor in a clathrin-, caveolae- and dynamin-independent, but lipid raft-dependent manner [43]. Once internalized, α-Syn activates microglia and triggers ROS generation through the NADPH oxidase system. In line with this, α-Syn-induced microglial activation was attenuated in mice lacking NADPH oxidase [42].

Proinflammatory cytokines and chemokines amplify the immune response and may be directly involved in neuronal cell death. Thus, TNFα induces apoptosis through the TNFR1 receptor death domain, which leads to the activation of caspases 1 and 3 [44]. Additionally, TNFα inhibits c-Rel, a NF-κB isoform that has a neuroprotective role as it inhibits apoptosis in dopaminergic neurons of the SN [45]. Indeed, NF-κB/c-Rel deficiency causes PD-like prodromal symptoms and progressive pathology in mice [46]. Moreover, high expression of the chemokine receptor CXCR4 and its ligand CXCL12 are observed in the SN of PD patients. CXCR4-CXCL12 signaling activates caspase 3, which in turn results in apoptotic neural death [47]. Binding of IFNɣ to its receptor IFNGR results in the phosphorylation of leucine-rich repeat kinase 2 (LRRK2) [48]. This kinase performs different functions both in microglia and dopaminergic neurons. Activated LRRK2 downregulates the expression of c-Rel [49], promotes the formation of TAU oligomers, which induces cell death [50], and favors an increase in membrane receptors of inflammatory cytokines [51]. Dopaminergic cell death also increases the exposure of surrounding cells to cellular debris and hidden proteins that promote immune cell activation [52].

Communication between different cell types is critical to properly resolve the inflammatory response. In fact, it has been suggested that the chemokine CX3CL1 (also called fractalkine) limits disproportionate inflammation in different proteinopathy models. This molecule is secreted by disease neurons as a “help me” signal, and through binding to its microglial CX3CR1 receptor, inhibits the production of proinflammatory cytokines and the expression of NOS2 [53]. Although its role in PD has not been fully elucidated, *Cx3cr1*-knockout mice display exacerbated neurodegeneration due to microglial dysregulation in the SN after MPTP exposure [54] and by intranigral injection of adeno-associated viral vectors overexpressing wild type or mutated α-Syn [55]. Moreover, overexpression of CX3CL1 attenuated the loss of dopaminergic neurons caused by α-Syn [56]. One of the mechanisms of action might be targeting the nuclear factor (erythroid-derived 2)-like 2 (NRF2) transcription factor [57], which counteracts inflammation [58].

Altogether, chronically activated microglia secrete high levels of proinflammatory mediators which damage neurons and further activate microglia, resulting in a feed-forward cycle promoting inflammation and neurodegeneration. Figure 1 illustrates the concept of the vicious cycle of neurodegeneration/neuroinflammation.

## 3. Role of Astrocytes

Astrocytes are the most abundant glial cell type in the CNS. Their cytoplasmic extensions connect directly with neurons and blood vessels in the neurovascular unit. Astrocytes metabolically support neurons by providing lactate for mitochondrial respiration, and are endowed with specific shuttle systems such as the malate–aspartate and glutamate–glutamine fluxes. Astrocytes also participate in tissue repair, filling the gaps left by dead neurons and secreting trophic factors required for neuron survival and synaptic function. Moreover, they participate in the maintenance and permeability of the BBB and cerebral blood flow [59].

There is evidence of astrogliopathy in the SN and striatum PD brains [60,61] and in animal models [37,62,63]. Reactive astrocytes experience gene expression changes as well as morphological rearrangement. Transcriptomics analyses have shown that different subpopulations of reactive astrocytes may exist depending on the surrounding microenvironment [64,65]. By analogy with microglia, the term A1 identifies astrocytes that resemble M1 proinflammatory microglia, and secrete IL1α, C1q and TNFα. These A1 cells lose their normal functions, including the nurturing of neurons, synapse function, and phagocytosis of altered synapses and myelin debris. Moreover, they secrete still unknown neurotoxic factor(s) that promote the death of neurons and oligodendrocytes. On the other side, A2 astrocytes, generated after insults, such as ischemia, upregulate many neurotrophic factors and are considered to be neuroprotective. Very importantly, A1 astrocytes are abundant in PD and appear to be activated, at least in part, through microglial interplay [65].

Amongst the 17 monogenic genes known to have a causative role in the development of PD, eight are expressed in astrocytes: *PARK7*, *SNCA*, *PLA2G6*, *ATP13A2*, *LRRK2*, *GBA*, *PINK1* and *PARK2* [66]. Some of these genes are closely involved in inflammation. For instance, DJ1 protein levels, encoded by *PARK7*, are higher in astrocytes than in neurons and this gene is upregulated in astrocytes from PD patients [67]. DJ1 regulates the assembly of lipid rafts involved in membrane receptor trafficking, endocytosis and signal transduction [68]. Therefore, mutations in *Park7* have been related to the disruption of the lipid rafts’ assembly. This fact not only causes impaired glutamate uptake, possibly leading to neuronal excitotoxicity, but also affects the immune responses mediated by astrocytes. Thus, DJ1 deficiency leads to impaired TLR3/4-mediated endocytosis [68] and altered production of proinflammatory factors [68,69,70].

In postmortem PD brains, α-Syn-positive inclusions have been detected not only in neurons, but also in astrocytes [61,71]. In fact, there is evidence of transmission of α-Syn from neurons to astrocytes [72] and the addition of extracellular α-Syn leads to accelerated production of proinflammatory cytokines (IL6, TNFα), increased expression of intercellular adhesion molecule 1 (ICAM1), and exacerbated generation of ROS [73,74].

In astrocytes, α-Syn localizes at the lysosome suggesting a role in autophagic degradation rather than in spreading along the brain parenchyma [75]. Intracellular α-Syn inclusions in astrocytes may affect the essential roles of these cells, such as uptake of glutamate or modulation of the BBB. Indeed, targeted astrocyte overexpression of mutated α-Syn in mice results in dopaminergic neuronal loss together with motor alterations. This effect is accompanied by astrogliosis and reduced expression of glutamate transporters [62].

Senescent astrocytes participate in PD development. When cultured human astrocytes or mice are exposed to the herbicide paraquat (PQ), associated with increased risk of developing PD, astrocytes become senescent both in vitro and in vivo and acquire a proinflammatory senescence-associated secretory phenotype [76]. This phenotype is characterized by the robust secretion of numerous proinflammatory cytokines, chemokines, growth factors and proteases [77]. Indeed, conditioned medium from senescent astrocytes compromised the viability of dopaminergic neurons. In line with this finding, the clearance of senescent astrocytes in PQ-treated mice diminished neurodegeneration. Interestingly, the authors showed that postmortem PD brain samples display increased astrocytic senescence (positive for p16INK4a, matrix metalloprotease 3 -MMP3- and the proinflammatory cytokines IL6, IL1α and IL8, and with reduced nuclear levels of LMNB1), strongly suggesting that senescent astrocytes contribute to the development of sporadic PD [76]. In fact, TNFα released by activated astrocytes bind to specific receptors expressed by dopaminergic neurons, such as TNFR1 and 2, and activate proapoptotic programs [78].

## 4. Endothelial Inflammation and Blood–Brain Barrier Permeability

Brain endothelial cells constitute a fundamental element of the neurovascular unit and accomplish numerous functions connected to the maintenance of CNS homeostasis. They regulate mechanotransduction, vascular permeability, rheology, thrombogenesis and leukocyte adhesion and traffic, concurrently ensuring optimal nutrient supply and protecting the brain from harmful insults [79].

Proinflammatory cytokines, such as IL1β or TNFα and other molecules released by activated microglia and astroglia, impact on endothelial cell functions [79]. Moreover, brain endothelial cells also express TLRs [80], and elicit an endothelium-derived inflammatory response to DAMPs and PAMPs. Activated endothelial cells produce proinflammatory cytokines (i.e., IL1β, IL6, TNFα), chemokines (i.e., CCL2, CXCL1) and adhesion molecules (i.e., intercellular adhesion molecule 1 -ICAM1-, vascular cell adhesion molecule 1-VCAM1, selectins), which participate in endothelial dysfunction. Under these conditions, they contribute to the recruitment of circulating blood cells into the brain microvasculature via the expression of adhesion molecules [79]. Interestingly, high levels of soluble VCAM1 were reported in plasma from PD patients when compared to age-matched controls, that correlated with disease stage and motor impairment [81], indicating strong blood–brain connections through cell traffic.

Vascular abnormalities in PD patients associated with neuroinflammation are accompanied by BBB leakage in the striatum [82] and in the midbrain [83]. Thus, positron emission tomography in PD patients has revealed a link between BBB permeability, infiltration of blood immune cells, such as CD4^+^ and CD8^+^ lymphocytes, and neuronal loss [84]. In the SN of PD patients, the vasculature presents a degenerated morphology characterized by the presence of endothelial “clusters,” due to vessel fragmentation and the loss of capillary connections [85]. In animal models, leakage of the BBB measured as elevated FITC-albumin content was observed in rats, 10 days after 6-OHDA unilateral injections in the striatum in correlation with a large number of activated microglial cells [86]. Some evidence pointing to inflammation as a primary culprit of alterations in BBB permeability stems from the observations that various cytokines including TNFα, IL1β or IFNγ decreased transendothelial electrical resistance in an in vitro model of the BBB [87]. Moreover, MPTP-induced BBB leakage was reduced in the *Tnfa*-knockout mice when compared to wild type littermates [88]. This was accompanied by reduced microglial activation and proinflammatory cytokine expression, but not by reduced neuronal loss, suggesting that the BBB dysfunction may be a consequence of neuroinflammation and not of cell death.

When considering vascular defects in PD, the impact of aging and the existence of several comorbidities in these patients must be considered [89]. Therefore, further studies will strengthen the above-mentioned observations and determine the impact of inflammation in PD-associated BBB dysfunction, in order to provide a better understanding of the disease etiology and also of novel therapeutic strategies.

## 5. Role of Peripheral Inflammation: Novel Hypothesis for PD Origin and Spreading

The neurovascular unit is altered in PD, allowing not only activation of the innate immune response, but also the recruitment and activation of the adaptive arm of the immune system [84,90]. It was hypothesized that oxidative modification of particular proteins associated with PD (i.e., α-Syn nitration) generates novel antigenic epitopes capable of initiating peripherally-driven CD4^+^ and CD8^+^ T cell responses [91]. Moreover, activated microglia induces the expression of MHC class I molecules in human catecholaminergic neurons, making them susceptible to cell death in the presence of cytotoxic T lymphocytes [92]. The importance of T lymphocytes was demonstrated in immunodeficient mice lacking mature T lymphocytes (*Rag1*^−/−^ and *Tcrb*^−/−^ mice), which displayed a remarkably reduced susceptibility to MPTP-induced dopaminergic neurodegeneration [84]. Moreover, systemic administration of IL1β led to exacerbated neuronal loss in the SN of 6-OHDA-treated rats probably by providing a costimulatory signal for the antigen-specific activation of T lymphocytes [93]. In fact, it has been shown that the chronic overexpression of a single proinflammatory cytokine in the SN, such as IL1β, can elicit most of the characteristics of PD, including progressive dopaminergic cell death, akinesia and glial activation [94].

It is noteworthy that increased cytokine levels, including IL1β, IL2, IL6, IFNγ, and TNFα, and higher CD4^+^ lymphocyte counts have been detected in serum and CSF from PD patients [95,96]. A peripheral proinflammatory phenotype is also supported by the finding of reduced CD4^+^ to CD8^+^ lymphocyte ratio and the number of Treg lymphocytes in patients vs. controls [97], supporting the idea that systemic inflammation is central to PD neurodegeneration. Peripheral inflammation has an echo at the brain level via BBB or the autonomic nervous system (the vagus nerve), and can switch the “primed” microglia into an “active” state that can trigger strong responses for sustaining the neurodegenerative processes [98].

The enteric nervous system and the immune system are tightly connected and influenced by gut microbiota and together constitute a field of intensive research due to its repercussion in PD (Figure 2). The intestine is enervated by nerve fibers of the sympathetic and parasympathetic system and, reciprocally, it affects the brain activity through the vagus nerve and the intestinal immune system [99]. Several events including activation of the vagus nerve increased in sympathetic nerve activity and intestinal microorganisms regulate this “microbiota–brain axis” and influence the intestinal and brain function [100]. In the context of PD, a relevant study [101] reported that dysregulation of the gut–brain interaction may explain the early gut disturbances in PD, and that nigral dopaminergic neurons exhibit increased vulnerability under the condition of gut inflammation. Thus, rats submitted to bilateral injection 6-OHDA exhibited in the colon an increase in inflammatory and oxidative stress markers, a decrease in types 1 and 2 dopamine receptors and an increase in the local dopamine levels. On the other hand, mice submitted to dextran sulfate-induced gut inflammation did not show significant changes in colonic α-Syn, but presented nigral dopaminergic neuron death, together with nigral proinflammatory markers and increased activity of the renin–angiotensin system.

Gastrointestinal physiology is influenced by signals generated in the brain and, in turn, alterations in the gut microbiota can affect brain neurochemistry by modifying not only the levels of neurotransmitters, hormones, neuropeptides and growth factors generated in the intestine [102], but also through the presence of circulating bacterial molecules that activate the NOD-like receptors (NLRs), providing a direct link between intestinal permeability and neuroinflammation [103,104]. Moreover, alterations in the gut microbiome have been described in PD patients [105,106,107], which also present intestinal inflammation and other anomalies [108,109]. In fact, fecal Calprotectin has been identified as a marker of activation of the gut immune system in PD patients [110]. Inflammatory bowel disease represents a risk factor to develop PD and this risk is significantly reduced in patients receiving anti-inflammatory therapy [111,112]. Additionally, the presence of α-Syn inclusions in the enteric system and vagal nerves has been detected in the very early stages of PD [113]. This might correlate with gut inflammation and trigger the activation of innate immunity in the gastrointestinal tract [8]. Gut microbiota may control inflammation in two ways: (1) by producing a milieu of mediators that exert a direct effect in the host (such as short-chain fatty acids –SCFA-, neurotransmitters or bacterial peptidoglycans) [114]; and (2) by providing structures that mimic self-antigens and trigger activation of T cells [115]. Regarding SCFAs, Sampson and coworkers [116] found that dietary supplementation with these lipids reduced motor symptoms, microgliosis and α-Syn. On the other hand, they reported aggravated Parkinsonism when transplanting gut microbiota from PD patients, suggesting that SCFAs enhance inflammation in PD. Moreover, the study by Unger and colleagues showed reduced SCFA levels in a small cohort of PD patients when compared to age-matched individuals [107]. Hence, we suggest that an altered SCFAs composition rather than then overall SCFA levels may have an impact on PD inflammation. In animal models, germ depletion in α-Syn overexpressing mice (ASO mice) attenuated microglial activation, α-Syn accumulation and motor symptoms. This was restored with SCFA supplementation. Conversely, the Parkinsonian phenotype was aggravated with the transplantation of gut microbiota from PD patients [116]. The recently established connection between the gut microbiota and the BBB [117], as well as with microglial function [118] should be considered in regard to PD origin and progression.

The pattern of α-Syn accumulation led Braak and coworkers to postulate that aberrant accumulation of α-Syn starts at the intestine, decades before the onset of symptoms, and spreads through the vagus nerve to the brain, mimicking a prion-like disease [119] (Figure 2). This hypothesis is supported by the observation that human PD brain lysate or recombinant α-Syn injection into the intestinal tissue is enough to induce the pathology in the vagus nerve and the brainstem of healthy rodents [120]. A recent study has shown that injection of PFF-α-Syn in the gut induces endogenous α-Syn aggregation and dissemination through the vagus nerve [121], and vagotomized patients show a decreased risk of PD [122]. All in all, alterations in the gut microbiota would modify the levels of certain metabolites (such as SCFA) and neurotransmitters, leading to inflammation and α-Syn pathology that would spread to the CNS resulting in PD motor symptoms.

Neurotropic pathogens may gain access to the brain through the nose [123]. The involvement of the olfactory bulb in PD is supported by different observations, including that (1) around 90% of PD patients lose the sense of smell, even before motor symptoms appear [123,124]; (2) the total volume of glomeruli in the olfactory bulb is reduced to 50% in PD patients vs. controls [125] and there is significant neuronal loss in regions connected to olfactory structures [123]; (3) α-Syn detected in the olfactory bulb of PD patients [126]; and (4) α-Syn injected in the olfactory bulb exerts a prion-like response [127]. The olfactory mucosa is exposed for decades to environmental contaminants that may initiate chronic inflammation, induce aggregation of α-Syn, and cause damage and dysfunctions that accumulate over time. So far, only one study has demonstrated microgliosis in the olfactory bulb in PD patients [128] and further research will definitely determine the impact of neuroinflammation provoked by this route. No correlation between nose microbiota and PD could be established, but the oral microbiome deserves additional research regarding its connection to PD [129].

Local neuroinflammation also has an impact on the peripheral immune system. Thus, in the murine model of intrastriatal injection of preformed fibril (PFF) α-Syn, in addition to microglial and astrocyte activation, there is significant infiltration of B, CD4+ T, CD8+ T, and natural killer cells. These penetrating T cells activate the microglial M1 proinflammatory phenotype, further spreading the neurodegenerative process [130]. In this model, mice display significant alterations in the composition of leukocyte subsets in the spleen and lymph nodes. Therefore, intracerebral-initiated synucleinopathies alter immune cell profiles both in the brain and peripheral lymphoid organs [41].

These facts considered, neuroinflammation can be regarded as the consequence of complex signaling between blood, gut and CNS. Further research is necessary to unravel the involvement of the peripheral immune system in disease initiation and propagation towards the CNS.

## 6. Genomics of PD-Related Inflammation

Some of the 17 genes related to the familial transmission of PD participate in inflammation. A very interesting finding is that the E3 ubiquitin ligase Parkin (*PARK6*), and the ubiquitin kinase PINK1 (*PARK7*) mitigate the inflammatory response to mitochondria-released DAMPs [131]. Parkin and PINK1 participate in the same biochemical pathway and remove damaged mitochondria via mitophagy. Genetic mutations in these two genes cause a recessive form of PD. Sliter and coworkers found that inflammation resulting from either exhaustive exercise or mtDNA mutation is exacerbated in *Prkn*^-/-^ and *Pink1*^-/-^ mice and this effect is completely rescued by concurrent loss of STING, a central regulator of the type I interferon response to cytosolic DNA. The authors also report that the loss of dopaminergic neurons from the SN and the motor alterations observed in aged *Prkn*^-/-^ mice are also rescued by loss of STING, suggesting the importance of inflammation at least in this type of PD. These results support a role in PINK1- and Parkin-mediated mitophagy in restraining innate immunity [131].

Mutations in *LRRK2* (*PARK8*) gene are the most common genetic cause of both familial and sporadic PD [132]. Besides PD, mutations in the LRRK2 gene are also found in immune-related disorders, such as inflammatory bowel disease (IBD) and some bacterial infections [133]. In a genome-wide association study, it was also reported that LRRK2 mutations are associated with autoimmune diseases such as type 1 diabetes, Crohn disease, ulcerative colitis, rheumatoid arthritis, celiac disease, psoriasis, and multiple sclerosis [134]. Therefore, LRRK2 seems to participate in the regulation of inflammatory responses not only in the brain but also at a systemic level. In fact, LRRK2 expression is expressed in microglia but also in peripheral immune cells such as monocytes, neutrophils, dendritic cells and, to a lower extent, in B and T cells [133,135]. LRRK2 modulates several pathways in innate and adaptive immunity by interfering with NF-κB and NF-AT signaling [136]. Several reports have shown increased expression of LRRK2 in immune cells challenged by particular stimuli such as IFNγ, LPS and IL1β [137], and influence both primary monocytes and microglia [138]. It has been suggested that PD-associated *LRRK2* mutations enhance LRRK2 levels in response to inflammatory stimuli. In turn, *LRRK2* deletion or pharmacological inhibition in rodent models prevent α-Syn-mediated neurodegeneration, and reduce inflammation and recruitment/activation of microglia [138,139]. Moreover, there is a positive correlation between LRRK2 protein levels in cytokine secretion and T cell activation that supports an immunoregulatory role of LRRK2 in PD [135]. Additional research is required to further understand the exact involvement of pleiotropic LRRK2 in PD-related immunity.

Genome-wide association studies (GWAS) have identified genetic variants that support an immune-related genetic susceptibility to PD. These genes are grouped into the functional categories of “regulation of lymphocyte activity” and “cytokine-mediated signaling”. For instance, PD risk loci within key immune-associated genes include *BST1* (bone marrow stromal cell antigen 1), proposed to play a role in neutrophil adhesion and migration. Moreover, several risk variants belong to the human leukocyte antigen (HLA) region encoding MHC class II molecules such as variants of the *HLA-DRB6* and *HLA-DQA1* loci [140,141]. This can impact on antigen presentation [142] considering that α-Syn-derived fragments act as antigenic epitopes displayed by HLA receptors for recognition by T cells [143]. Therefore, the MHC-II locus is seemingly linking environmental factors to genetic susceptibility in conferring risk for PD.

## 7. Transcriptional Control of Inflammation in PD

Nuclear factor-kappa B (NF-κB) comprises seven isoforms that regulate survival and inflammatory responses in various cell types. The p65-NF-κB component is considered the master regulator of inflammation. Briefly, NF-κB is retained in the cytosol through interaction with the nuclear *κ* B inhibitor α (I*κ*Bα) in resting immune cells. Meanwhile, PAMPs and DAMPs activate cognate receptors in immune cells that result in the activation of the I*κ*B kinase. This kinase phosphorylates I*κ*B, targeting it for degradation and allowing nuclear translocation and activation of the NF-κB transcriptional program [144]. NF-κB is activated in the SN of PD patients as well as in MPTP-treated mice [145] and the hemiparkinsonian monkey [146]. Since this transcription factor plays a key role in controlling the expression of proinflammatory mediators such as IL1β, IL6, TNFα and COX2, these observations are in line with the already mentioned increase in the levels of these molecules in patients and in preclinical models [147]. Congruently, NF-κB inhibition suppresses proinflammatory cytokine expression, protecting dopaminergic neurons and improving motor activity [145] (Figure 3A).

Nuclear receptor related 1 protein (NURR1) is a member of the NR4A family of orphan nuclear receptors [148] with a key role in the generation, development, maturation and maintenance of dopaminergic neurons [149]. Mutations in the gene encoding NURR1 are associated with familiar cases of PD and decreased expression of this transcription factor has been detected in PD brains [150,151]. Interestingly, in addition to dopaminergic neurons, NURR1 is also expressed in microglia and astrocytes, where it inhibits the generation of proinflammatory mediators and consequently protects against inflammation-mediated neuronal death. Specifically, it binds to NF-κB responsive elements in the promoter region of proinflammatory genes and recruits the CoREST corepressor complex, thereby reducing the expression of these genes [152] (Figure 3B).

PPARS have emerged as links between lipids, metabolic diseases and innate immunity [153]. Although the three isoforms of PPAR (PPARα, PPARβ/δ and PPARɣ) are expressed in neurons and glia, PPARβ/δ is the prevalent isoform found in the brain [154], that responds to saturated and unsaturated fatty acids, metabolites of the arachidonic acid generated by lipoxygenases (15-HETE), and particular components of the plasma lipoprotein fraction [155]. In the MPTP mouse model of PD it has been shown that NLRP3 inflammasome-mediated neuroinflammation plays a critical role in the pathogenesis of PD [156], and that the specific PPARβ/δ agonist GW501516 can alleviate neuroinflammation in astrocytes [157]. Moreover, GW501516 conferred neuroprotection in the rotenone rat model of PD by suppressing the IRE1α-caspase-12-mediated ER stress pathway [158]. While the PPARγ agonist rosiglitazone prevents LPS-induced microglial activation, the PPARγ antagonist T0070907 induces the M1-to-M2 phenotypic shift in activated microglia, to inhibit NF-κB activation and to promote microglial autophagy [159]. PPARs also repress genes implicated in the inflammatory response by interfering with several inflammatory pathways, such as the inter-related NF-κB, AP1 or NF-AT signaling pathways that are critically involved in neurodegenerative diseases [160]. As reviewed by Le Menn and Neels [161], the anti-inflammatory action of PPARs is underlined by: (a) transrepression mechanisms involving NF-κB and AP-1 in macrophages; (b) inhibition of inflammatory Th1 and Th17 polarization, and shifting of CD4+ T cells towards the Th2 phenotype. Moreover, it has been shown that pioglitazone, another PPARγ agonist, inhibits microglial activation, reduces the production of proinflammatory factors and protects dopaminergic neurons, interfering with phosphorylation of JUN and NF-κB, complemented by the suppression of the inducible COX2 expression and the subsequent PGE_2_ synthesis [162].

PPARs form heterodimers with the retinoid X receptor (RXR) for binding to PPAR responsive elements located in the promoter of target genes [163]. Neurons and glial cells in the SN contain retinoic acid receptors (RARs), and dopaminergic neurons express high levels of the enzymes necessary to convert vitamin A into retinoic acid (RA). Several reports point to the importance of RA in the maintenance of the nigrostriatal pathway [164]. In fact, reduced levels of RALDH1, an enzyme involved in RA biosynthesis, were observed in the surviving neurons of PD patients [165]. Interestingly, RA seems to promote anti-inflammatory responses in the CNS. For instance, the treatment of astrocytes with RA prevented LPS-induced secretion of proinflammatory cytokines IL1β, IL6 or TNFα [166]. Therefore, reduced RADHL1 levels can lead to decreased RA in the midbrain, making patients more susceptible to proinflammatory processes. In this scenario, targeting RARs with synthetic RA would help counteract inflammation and preserve the dopaminergic neuronal population (Figure 3C).

Nuclear factor (erythroid-derived 2)-like 2 (NRF2) belongs to the cap’n’collar b-Zip family of transcription factors. Although initially described as a critical regulator of the antioxidant response, NRF2 is currently viewed as the master regulator of cytoprotective responses. This transcription factor controls the expression of genes involved not only in the maintenance of redox balance, but also in metabolic pathways, regulation of proteostasis and resolution of inflammation. In this regard, NRF2 acts as a brake in the inflammatory response by different means, including the modulation of redox metabolism, antagonizing NF-κB or directly inhibiting the expression of proinflammatory cytokines [58] (Figure 3D). Several reports have shown that pharmacological activation of NRF2 has beneficial effects in animal models of PD [167]. For instance, in the murine MPTP model, NRF2 deficiency led to more severe dopaminergic dysfunction accompanied by exacerbated astrogliosis and microgliosis [168]. The protective role of NRF2 observed in animal models is consistent with the observation that certain single nucleotide polymorphisms in the promoter of the gene encoding NRF2 (*NFE2L2*) conform a protective haplotype, delaying or even reducing the risk of PD [169,170]. Recently, mutant α-Syn was demonstrated to inhibit the NRF2-mediated antioxidant response, leading to axonal pathology through the loss of expression of microtubule-stabilizing proteins [171].

## 8. The Immunomodulatory Role of Dopamine

Neurotransmitters, in particular dopamine, not only mediate neuronal communication, but also crosstalk between the nervous and immune systems [172]. Dopamine is produced in the peripheral and the central nervous system, and is generally associated with reward-motivated behavior in the brain. Lymphoid tissues can be innervated by dopaminergic neurons [173] and many types of immune cells (T and B lymphocytes, dendritic cells, monocytes/macrophages, neutrophils and natural killer cells), respond to dopamine via specific receptors expressed in both the innate and adaptive immune systems [174]. These receptors show distinct affinities for dopamine and, consequently, this neurotransmitter displays complex immunomodulatory roles, depending on its concentration, the receptor subtype and the target immune cell.

Central dopaminergic hypoactivity has been related to increased risk of inflammation [175]. For instance, dopaminergic inhibition of the immunosuppressive Treg function in response to endogenous or exogenous dopamine [176] was shown to occur through the activation of D_1_-like dopamine receptors that activate adenylate cyclase [177], hence favoring the uncontrolled proliferation of effector T cells [174]. Moreover, dopamine may differentially drive the differentiation of CD4^+^ T cells into Th1 or Th17 inflammatory phenotypes, depending on the context, and this could represent a driving force during autoimmunity [176]. In turn, dopamine can limit inflammatory processes by exerting an inhibitory effect on the NLRP3 inflammasome [178]. Thus, the dopamine D2 receptor inhibits NLRP3 inflammasome activation in astrocytes through a β-arrestin2-dependent mechanism [179].

It is worth noticing that immune cells, such as antigen-presenting cells (dendritic cells, monocytes), can produce by themselves dopamine and respond in an autocrine manner to dopamine or release it for modulating neighboring immune cells [176]. For instance, dopamine stored in human monocyte-derived dendritic cells can be released in the immunological synapse [180] after antigen-T cell interaction, and promotes T cell differentiation towards the anti-inflammatory Th2 functional phenotype [181]. In agreement with reduced dopaminergic activity in PD, patients present reduced proportions of Th2 cells [182].

As already mentioned, CD4^+^ T-cell infiltration into the SN has been demonstrated in both patients and animal models. Thus, deficiency of the dopamine receptor D3 in CD4^+^ T cells renders MPTP-treated mice resistant to microglial activation and degeneration of nigral dopaminergic neurons [183]. Similar results have been reported in astrocytes expressing the D3 receptor [184].

While the role of dopamine in systemic inflammation is widely recognized, only recently it was shown that dopamine inhibits TRAF6-mediated NF-κB activation and inflammation via the D5 dopamine receptor in macrophages [185]. Through this mechanism, dopamine protected mice against *S. aureus*-induced sepsis and meningitis. In another recent study, it was reported that myeloid-specific dopamine D2 receptor signaling controls inflammation in acute pancreatitis via inhibiting M1 macrophage polarization [186].

Dopamine can be also produced by neurons from the enteric system [187] and peripheral dopamine levels may be influenced, thus, by gut microbiota. It is plausible to think that alterations in gut microbiota or α-Syn aggregation lead to disturbances in the enteric system, altering intestine permeability and leading to early gut inflammation. In addition, local changes in the activity of enteric dopaminergic neurons affect the autonomous nervous system and together with alterations in other catecholaminergic neurotransmitters may lead to some dysautonomias of PD, including neurocardiological and other nonmotor abnormalities [188,189].

## 9. Oxidative Stress and Neuroinflammation: A Vicious Circle

There is a strong correlation between PD, redox imbalance, and low-grade chronic inflammation. Elevated levels of several of inflammatory mediators (TNFα, NO, PGE_2_, IL1β, IL6) and oxidized biomolecules (4-hydroxynonenal, oxidative cholesterol metabolites, 8-oxoG) are found in the cerebrospinal fluid of PD patients, as well as in postmortem samples of SN of PD patients [190,191,192]. Dopaminergic neurons are, therefore, exposed to high levels of oxygen and nitrogen reactive species (ROS and RNS, respectively). Both species are generated from different sources including dopamine degradation or autooxidation, iron-binding to neuromelanin, mitochondrial electron transport chain [193] and loss of proteostasis due to reduced degradation capacity, which will favor the aggregation and oxidative modification of several proteins, including α-Syn [167]. Despite these detrimental effects, ROS play a role in phagocytic cells acting as microbicides as well as signaling molecules. In microglia, ROS are mainly produced by the multi-subunit enzyme NADPH oxidase (NOX) [194]. NOX transfers electrons from NADPH and NADH [195] to molecular oxygen generating superoxide anion. In surveilling microglia, p40phox, p47phox and p67phox are in the cytosol, separated from the rest of the NOX-subunits (cytochrome b558, comprising p22phox and gp91phox). Upon stimulation, all the subunits are assembled in the membrane activating the catalytic capacity of NOX [196]. ROS regulate signaling through the oxidation of low pKa cysteine residues that exist as thiolate anions at physiological pH, rendering them susceptible to oxidation by ROS. Oxidation of such redox-sensitive cysteines typically leads to the inhibition of the enzymatic activity of targeted proteins e.g., protein tyrosine phosphatases, the lipid phosphatase PTEN, and regulatory enzymes of ubiquitin and ubiquitin-like proteins such as SUMO and Nedd8 [197]. This system enables rapid regulation of downstream signaling pathways, transcriptional regulation, and activation of the inflammatory response. However, ROS are recognized as a double-edged sword since they are highly reactive and produce molecular damage in DNA, protein, and lipids, resulting, for example, in DNA strand breaks, irreversible modification of protein tyrosine residues, and loss of membrane integrity.

In the PD brain, the sustained activity of microglial NOX might exert pathological effects both by direct ROS damage to neighboring neurons and by triggering inflammatory cytokine signaling that results in a vicious circle of neuronal damage. Microglial cells intensely express gp91phox and p47phox subunits [198,199]. In line with these findings, the degeneration of DA neurons induced by MPTP was attenuated in gp91phox-null mice compared with wildtype littermates [200]. There are seven isoforms of NOX, being NOX1 and NOX2 especially relevant in PD. Both isoforms are increased in the SN of PD patients, suggesting that the NOX plays a role in the degeneration of those neurons [201,202]. NOX2 expression levels are increased in M1- compared with surveilling microglia in both humans [203] and mice [204]. Aggregates of α-Syn stimulate NOX2, eventually contributing to dopaminergic damage [42]. Recently, it has been proposed that blockade of CD11b, the α chain of integrin αMβ2, but not TLR2, attenuated α-Syn-induced NOX2 activation in microglia [205]. Mutant mice defective in NADPH oxidase exhibit less dopaminergic neuron loss and protein oxidation than their wildtype littermates after MPTP treatment [201]. Similar approaches involve NOX1 in the 6-OHDA [206] and in the paraquat [207] models of PD.

Another hallmark of the proinflammatory phenotype is the generation of NO by microglia. NO is mainly synthesized from L-arginine and O_2_, not only by the inducible NOS (NOS2) isoform, which is independent of calcium rise, unlike the other two constitutive NOS isoenzymes (endothelial NOS -NOS3-, and neuronal NOS -NOS1-) that also contribute to NO production although at moderate concentrations. NOS2 is transcriptionally induced by inflammatory mediators such as LPS and proinflammatory cytokines in many mammalian species but, importantly, the expression in primates, in particular in humans is very restricted, playing a key role the NO produced by the constitutive NOS [22]. Recruitment of the transcription factor ΝF-κB by activation of different signaling pathways including JAK, SRC family, MAPKK, protein kinase A, phosphatidyl inositol-3 kinase, and protein kinase C is crucial to activating NOS2 expression [208]. Very relevant, there is a connection between NOS and NOX systems since NOS2 expression is also regulated by intracellular ROS production [208]. In fact, microglial cells submitted to either antioxidant compounds or NOX inhibitors reduce their NO content [209]. Due to its reactivity with ROS, toxic RNS species could be generated. NO reacts with superoxide forming peroxynitrite (ONO_2_^−^), one of the most potent oxidant molecules released by living cells that irreversibly modifies thiols and amines by nitrosylation and nitration reactions. Therefore, synergic activation of NOX and NOS negatively impacts neuronal survival in response to different proinflammatory stimuli, such as TNF, IL1β, LPS, ATP, or phorbol 12-myristate 13-acetate [210].

Two transcription factors that are important in the regulation of microglial fate are ΝF-κB and NRF2. The equilibrium between both factors is central to define the phenotype polarization of these cells. After an inflammatory challenge, microglia execute an M1 program and ROS/RNS increase the phosphorylation levels of the kinases that control ΝF-κB to further upregulate the proinflammatory phenotype. NF-κB inhibitors impair the acquisition of the M1 phenotype since they reduce TNFα, IL1β, and IL6, as well as ROS/RNS levels [211].

In response to the same rise in ROS/RNS levels, but probably in a retarded fashion, NRF2 levels increase, favoring M2 conversion to restore redox homeostasis and prevent inflammation over-load. Accordingly, NRF2-deficient mice exhibited increased levels of COX-2, NOS2, IL6, and TNF (M1 markers) and reduced levels of FIZZ-1, YM-1, ARG, and IL4 (M2 markers) in response to MPTP compared to wild type littermates [168,212] (Figure 4).

Astrocytes also contribute to inflammation and oxidative stress in the PD-brain. Resting astrocytes become A1-astrocytes in response to IL1α, TNF and C1q secreted by M1-microglia. A1 astrocytes do not reinforce neuronal survival, outgrowth, or synaptogenesis and, instead, induce neuronal death [65]. The molecular mechanisms are not fully described but A1-astrocytes generate TNFα, IL1β, IL6, and IFNγ, which initiates neuronal apoptosis through the activation of caspase 3, caspase 8, and cytochrome c release to the cytosol [213,214]. In addition, NO release has been also depicted in the extracellular space by A1, contributing to oxidative damage.

Astrocytes can take α-Syn from neurons through endocytosis and form inclusion bodies. Accumulation of α-Syn in astrocytes promotes a proinflammatory profile. Hence, the induction of proinflammatory cytokines and chemokines correlated with the extent of glial accumulation of α-Syn [72]. Oligomeric α-Syn significantly increases the rate of production of ROS and lipid peroxidation in primary astrocytes [215]. TLR4-null astroglia present a suppressed proinflammatory response and decreased ROS production in response to in vitro treatment with α-Syn [216]. Very relevant, dopaminergic damage can be ameliorated by NLY01, an agonist of the glucagon-like peptide-1 receptor, which prevents the conversion into A1-phenotype [217]. On the other hand, astrocytes play a critical role in maintaining the CNS antioxidant system and neutralizing ROS/RNS, as they release reduced glutathione (GSH) and superoxide dismutase to the microenvironment [218] and provide precursors of GSH synthesis in neurons [219]. Consistent with the loss of astrocytic physiological functions in PD, decreased GSH levels are observed in the SN [220]. Recently, it has been shown that GSH depletion with erastin, or a direct GPX4 inhibitor RSL3, ultimately induces intensive lipid peroxidation that causes cell death by ferroptosis [221].

To avoid the detrimental effects of ROS/RNS brain cells display an armamentarium of antioxidant enzymes, some of them controlled by the transcription factor NRF2. The role of NRF2 in glial cells is expected to be especially relevant as they show the highest expression levels in the brain [222]. In fact, astrocytes are highly responsive to NRF2 activation in several models of neurodegeneration [223] and as discussed above microglia from *Nrf2*-null mice exhibited higher levels of proinflammatory mediators in response to MPTP [168,212]. Interestingly, several NRF2-target genes, including *HMOX1*, *NQO1*, *GCLM* and *SQSTM1*, are upregulated PD brains [224,225,226]. Genetic analyses between PD risk and the functional haplotype made of SNPs in the *NFE2L2* promoter have demonstrated the relevance of this upregulation. Data from Swedish, Polish and four independent European cohorts associated this haplotype with delayed onset and reduced risk of PD [169]. Afterwards, these findings were not replicated in a Taiwanese population [170], suggesting differences in ethnicities and/or environmental factors.

Taken together, these findings demonstrate that sustained activation of microglial and astroglia cells is associated with PD by directly contributing to PD pathogenesis or because of its progression.

## 10. Therapeutic Strategies Targeting Neuroinflammation in PD

Current therapies for PD turn around maintenance of dopamine levels either by inhibiting degradation of endogenous dopamine (monoamine oxidase B and catechol-O-methyl transferase inhibitors) or by supplying the dopamine precursor levodopa or dopamine agonists. However, these therapies are only symptomatic. Given that inflammation is a crucial pathomechanism in PD, immunomodulatory therapies are a promising field for research to endorse a neuroprotective strategy. Epidemiological studies aimed at determining if nonsteroidal anti-inflammatory drugs reduce the risk of PD have not shown consistent results, possibly due to methodological as well as chemical and functional differences of the analyzed drugs [227]. However, other alternatives are being tested as discussed below for preclinical (Table 1) and clinical studies. (Table 2)

Several preclinical and clinical studies have tried to inhibit the glial reaction and/or target proinflammatory cytokines with minocycline, dexamethasone or COX2 inhibitors [228,229,230,231] (NCT01439100, NCT00063193). A recent study has demonstrated that the NLRP3 inhibitor, MCC950, effectively suppressed microglial inflammasome activation, mitigating motor deficits, nigrostriatal dopaminergic degeneration and accumulation of α-Syn aggregates in multiple PD models [232]. These findings highlight NLRP3 as a potential therapeutic disease-modifying target for PD.

Direct inhibition of microglial activation or IL1- and TNF-neutralizing antibodies (already approved by the FDA) could also prevent the acquisition of the A1 astrocytic phenotype. Anti-TNF therapy, tested in patients suffering from inflammatory bowel disease, was associated with substantially reduced PD incidence [112]. Recently, it has been demonstrated that the glucagon-like peptide 1 receptor (GLP1R) agonist NLY01 protects dopaminergic neurons inflammatory responses elicited by astrocytes and microglia [217]. Importantly, there is a clinical trial ongoing to evaluate the neuroprotective and anti-inflammatory properties of a synthetic analog of GLP1, semaglutide, in idiopathic PD (NCT03659682). Another strategy to slow down PD progression would be to promote the resolution of inflammation. In this regard, resolvin D1 (a proresolving molecule) prevented central and peripheral inflammation, together with neuronal dysfunction and motor deficits in rats expressing α-Syn [233].

Strategies targeting the peripheral immune system have been designed to prime T cells with different agents and transfer them to the periphery of animal models with induced dopaminergic neuron death. For instance, adoptive transfer of T cells immunized with glatiramer acetate (a synthetic random amino acid polymer used as an immunization-based antigen) to MPTP-treated mice led to the infiltration of T cells in the SN, suppressed microglial activation, and increased the synthesis of astrocyte-derived NFs, resulting in protection of dopaminergic neurons [234].

The transmissible nature of α-Syn underlying the progression of PD has opened new possibilities for therapeutic strategies. It is possible that inflammation promotes the prion-like transfer of α-Syn by increasing its release, its uptake, or both. Therefore, reducing the underlying inflammation and/or the cell-to-cell α-Syn transfer could be a therapeutic strategy to slow down the progression of PD. Two immunotherapeutic strategies have been explored so far: (1) active immunization using the patient’s own immune system to generate antibodies against α-Syn or (2) passive immunization using the direct administration of antibodies against different domains of α-Syn [239]. In the first case, a vaccination-based approach called AFFITOPE^®^ AFF1 based on the administration of short fragments of α-Syn conjugated to a carrier stimulated the generation of antibodies against α-Syn in two mouse models of PD, reduced neuropathology and increased anti-inflammatory cytokine expression [236]. Based on the preclinical results, this strategy moved to a phase I clinical trial for PD, showing good levels of immunogenicity (NCT02267434). On the other hand, passive immunotherapy, mostly directed to the α-Syn carboxi-terminus, has also shown beneficial results in preclinical studies, presumably through augmenting α-Syn clearance by microglia [240,241]. A phase 2 clinical trial was performed to evaluate the humanized form of 9E4 (PRX002) and assess safety and pharmacokinetics in patients with idiopathic PD (NCT03100149).

The involvement of gut microbiota in the initiation of chronic inflammation and α-Syn aggregation in the enteric system provides new therapeutic opportunities that have been poorly analyzed so far. In this regard, it was demonstrated that the consumption of fermented milk containing multiple probiotic strains and prebiotic fiber improved constipation in patients with PD [242]. Additional studies must be conducted in order to determine whether this approach represents a disease-modifying therapy for PD. Interestingly, an ongoing clinical trial will determine whether the antibiotic rifaximin restores the gut microbiota in people with PD and if this is associated with the reduction of systemic inflammation and α-Syn (NCT03958708). Another study has been designed to evaluate fecal transplantation in gut inflammation in PD (NCT03808389).

Transcriptional programs that modulate inflammation, such as transcription factor NRF2 and PPARγ, have also been the focus of intensive research. Activators of NRF2 not only reduce the inflammatory process but also the oxidative burden and the proteinopathy associated with PD [167,222]. In this regard, repurposing dimethyl fumarate, already approved for the treatment of multiple sclerosis, has shown promising results in preclinical models of PD [224].

Some PPARγ agonists currently used in the clinic for type 2 diabetes mellitus have been re-evaluated in animal models of PD. Besides their well-known regulatory role on lipid and glucose metabolism, PPAR-γ agonists also exhibit anti-inflammatory properties [245]. The repurposing of these drugs for PD is highly justified by the fact that diabetes is a frequent comorbidity of PD. Thus, initial studies conducted with 51,552 Finish [246], 21,841 US [247] and 288,662 US [248] subjects indicated a correlation between type 2 diabetes and the risk of Parkinson’s disease. In a Danish study, diabetes was associated with a 36% increased risk of developing Parkinson’s disease [249]. Moreover, the progression of PD is worse for individuals who are diabetic before the time of diagnosis [250]. Thus, pioglitazone prevented neuronal death and microglial activation in several murine and nonhuman primate models of PD [237,251]. Additionally, the chronic administration of rosiglitazone attenuated behavioral deficits, dopamine depletion, dopaminergic cell death and microglial activation in the MPTP mouse model of PD [238,252]. Although a clinical trial reported no benefits in human patients [253], some retrospective studies support reduced PD risk in diabetic patients taking glitazones [243,244].

Since inflammation is considered an early event in PD, targeting inflammatory pathways might be more effective in preventing disease progression than in reversing existing pathology. Therefore, efficient therapies are dependent on the improvement of early diagnosis. In this regard, specific analysis of combined inflammatory components might provide useful biomarkers for monitoring the progression of the disease [254,255]. Further research and ongoing clinical trials (NCT03633513; NCT03716258) will warrant the specificity and usefulness of inflammatory/immunological biomarkers for PD.

PD is a chronic disease, making it likely that prevention and treatment will require long-term therapy. Thus, a major concern of targeting inflammatory processes is the risk of immunosuppression, enabling opportunistic infections. Therefore, high selectivity must be a requisite for this therapy. Furthermore, due to the complex nature of the disease and the number of cell types involved, it may take more than one drug to confer a therapeutic benefit. Therefore, future strategies should try to consider ‘systemic protection’ instead of just single ‘neuroprotection’.

## 11. Concluding Remarks

Chronic inflammation can directly or indirectly contribute to the etiology and progression of PD. A possible scenario is that chronic inflammation in the olfactory tract and/or the enteric system due to environmental and pathogen exposure throughout life and/or alterations in gut microbiota favors α-Syn aggregation and spreading to the brain, further aggravating inflammatory responses that contribute to neurodegeneration. Therapeutic intervention for PD remains an urgent challenge. Due to the central role of inflammation in PD, immunomodulatory therapies become prime targets for research. Although inhibition of neuroinflammation itself, either independently or as a combined strategy to other interventions, may not alter the underlying cause of the disease, it may reduce the production of factors that contribute to neurotoxicity, thereby resulting in clinical benefit.

## Figures and Tables

**Figure 1 cells-09-01687-f001:**
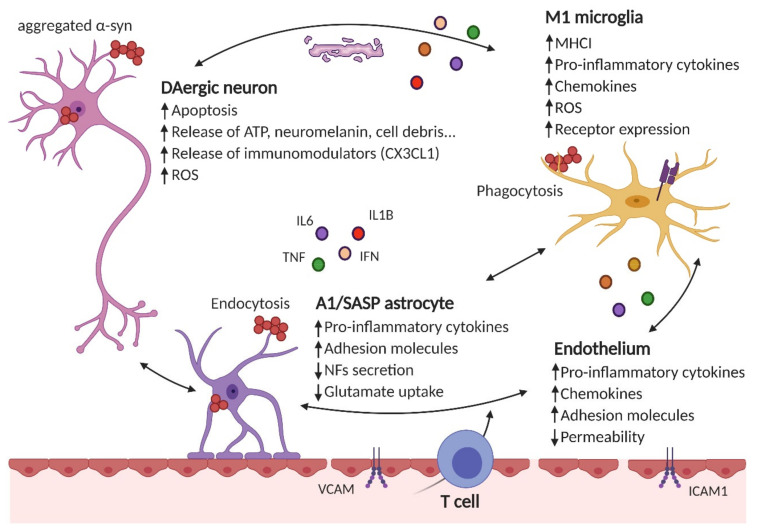
Neuroinflammation results from the crosstalk between different cell types in the brain. Neurons, astrocytes, microglial or endothelial cells are susceptible to α-Syn aggregates (i.e., by phagocytosis, endocytosis, Toll-like receptor (TLR) stimulation, etc.), which can result in the impairment of their homeostatic functions (reduced secretion of neurotrofic factors -NFs-, impaired glutamate uptake, etc.) and secretion of proinflammatory cytokines (such as IL6, IL1β, TNFα, IFNγ, etc.), chemokines (CCL2, CXCL1, etc.) and increased receptor expression (for proinflammatory cytokines and chemokines, MHCI in microglial cells, adhesion molecules in the endothelium, etc.). Additionally, peripheral immune cells (such as CD4^+^ T cells) are recruited to the brain parenchyma. These immunomodulatory mediators and/or the lack of efficient resolving mechanisms further increase the proinflammatory environment.

**Figure 2 cells-09-01687-f002:**
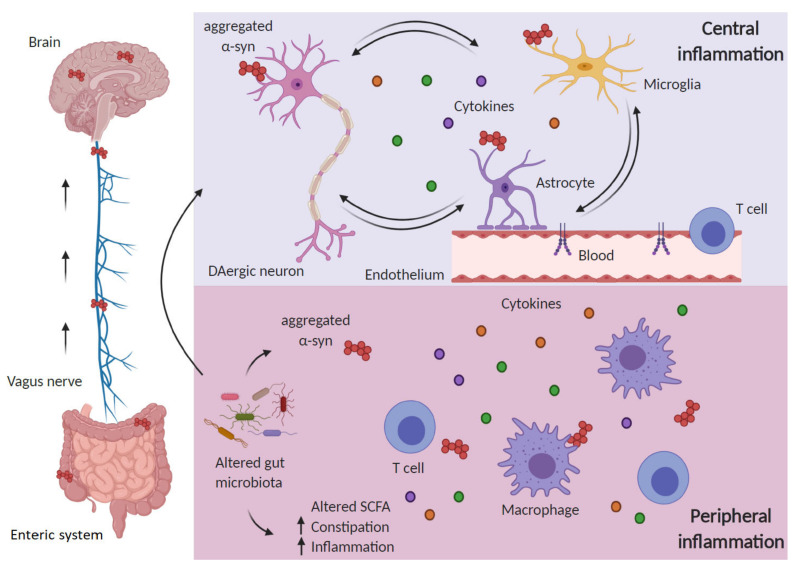
Inflammation in Parkinson’s disease (PD) encompasses central and peripheral inflammation. The “gut–brain axis” hypothesis in PD holds that alterations in the gut microbiota may favor α-Syn aggregation and are responsible for an inflammatory response in the periphery, which includes increased cytokine levels and activated T cells. Aggregated α-Syn is suggested to spread from the periphery to the brain through the vagus nerve in a prion-like manner. Once in the brain, proteinopathy together with other triggering factors (mitochondrial impairment, ROS, etc.) will sustain central inflammation in a vicious circle between dying dopaminergic neurons, glial cells and activated endothelium, further aggravated by infiltrating peripheral immune cells.

**Figure 3 cells-09-01687-f003:**
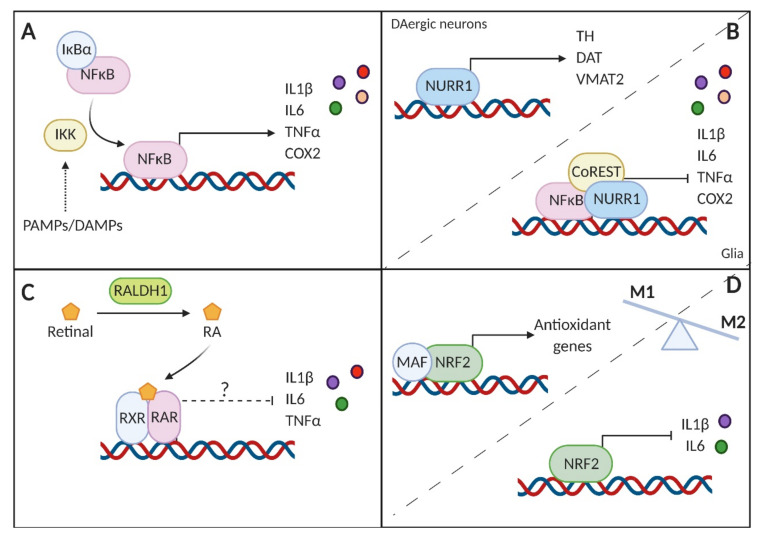
Schematic representation of the transcriptional modulation of the inflammatory response in PD. (**A**) Nuclear factor-kappa B (NF-κB) is subjected to tight regulation by the nuclear κ B inhibitor α (IκBα). Upon activation by pathogen-associated or damage-associated molecular patterns (PAMPs or DAMPs, respectively), the IκB kinase (IKK) targets IκBα for degradation, allowing NF-κB to transactivate the expression of proinflammatory genes. (**B**) Nuclear receptor related 1 protein (NURR1) controls the expression of essential genes for the survival of dopaminergic (DAergic) neurons, but also in glial cells has the potential to repress the activity of NF-κB when recruiting the corepressor complex CoREST. (**C**) Reduced levels of the enzyme RALDH1 in PD may impair proper synthesis of retinoic acid (RA), essential for the nigrostriatal pathway. Moreover, RA and its derivatives have shown to have key anti-inflammatory effects, that, therefore, would be lessened in PD. (**D**) NRF2 controls the transcription of antioxidant genes, thereby reducing the oxidative burst. NRF2 directly represses the transcription of the proinflammatory interleukin 1β (IL1β) and interleukin 6 (IL6). The overall effect of this transcription factor is to counteract the proinflammatory phenotype (M1) in favor of the anti-inflammatory (M2) phenotype.

**Figure 4 cells-09-01687-f004:**
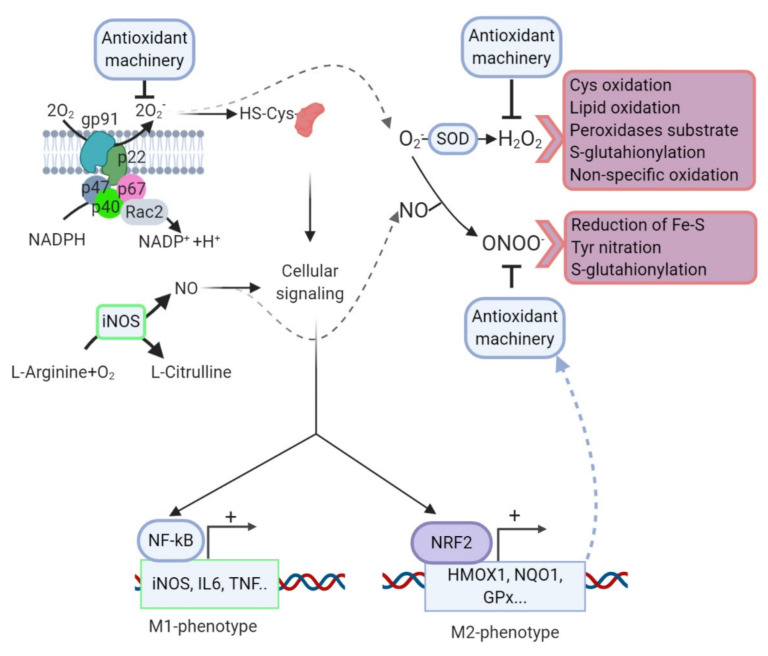
Redox control of microglial phenotype. Under surveillance mode, microglial cells exhibit low levels of ROS/RNS properly managed by the antioxidant machinery. After an inflammatory challenge, microglia activate an M1 program that is characterized through a rapid and high increase in ROS/RNS levels mainly derived from the NOX and NOS activities. During this phase, ROS/RNS act as second messengers increasing the phosphorylation levels of the kinases that control ΝF-κB to further upregulate the proinflammatory M1 gene profile. However, other transcription factors, including NRF2, are increased in response to ROS/RNS but probably in a second wave. NRF2 will restore redox homeostasis and attenuate M1 in favor of M2 phenotypes. Crosstalk between NOX and NOS systems, H_2_O_2_ generated after superoxide anion dismutation can induce cysteine oxidation, S-glutathionylation, lipid peroxidation, and a reaction with other peroxides or nonspecific oxidation of other molecules. NO, results of the transformation of L-arginine by NOS, can react with superoxide anion generating peroxynitrate, a molecule that can modify tyrosine residues by nitration, S-glutathionylation of diverse molecules, or reduction of ferric sulfide.

**Table 1 cells-09-01687-t001:** Some pre-clinical studies aimed to combat neuroinflammation in PD.

Molecular or Cellular Target	Pharmacological Intervention	Results	Reference
Glucocorticoid receptor	Dexamethasone	Prevention of the degenerative effect of intranigral LPS injection	[228]
Several G protein-coupled receptors	Resolvin D1	Protection against nigrostriatal overexpression of α-Syn	[233]
Immunomodulator	Minocycline	Antiapoptotic, anti-inflammatory, and antioxidant effects on several PD models	[229]
COX2	Nonsteroidal anti-inflammatory drugs	Prevention of inflammation and dyskinesia in the rotenone rat model	[235]
NLRP3	MCC950	Prevention of inflammation and dopaminergic death in mice submitted to preformed fibrils of α-Syn	[232]
T lymphocytes	Glatiramer acetate	Immunization strategy for suppression of microglial activation in (MPTP)-intoxicated mice	[234]
B lymphocytes	Short fragments of α-Syn	Vaccination against c-terminal fragment of α-Syn in murine models of PD and Dementia with Lewy bodies	[236]
NRF2	Dimethyl fumarate	Antioxidant and neuroprotective effects on the mouse model of nigrostriatal overexpression of α-Syn	[224]
PPARγ	Pioglitazone	Reduction of dopaminergic cell deadnadn microgliosis in the MPTP intoxicated *Reshus* monkeys	[237]
	Rosiglitazone	Improvement of motor activity and reduced neurodegeneration and inflammation in a chronic MPTP mouse model	[238]

**Table 2 cells-09-01687-t002:** Some clinical trials aimed to combat neuroinflammation in PD.

Molecular or Cellular Target	Pharmacological Intervention	Results	Reference
TNF	Anti-TNF antibodies Retrospective cohort study	A higher incidence of PD was observed among patients with inflammatory bowel disease	[112]
GLP1	Semaglutide (Phase 2) Not yet recruiting	neuroprotective and anti-inflammatory in idiopathic PD	NCT03659682
Gut microbiota	Rifaximin (Phase 2) Recruiting	Change of gut microbiota. Blood biomarkers of neuroinflammation and exosomal alpha-synuclein	NCT03958708
Gut microbiota	Fecal transplantation. Recruiting	Change in clinical symptoms	NCT03808389
PPARγ	Pioglitazone. Retrospective cohort study	lower incidence of PD among diabetic patients	[243,244]

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
