# Peer review of "Inflammation in Parkinson’s Disease: Mechanisms and Therapeutic Implications"

_cells, 2020, doi:10.3390/cells9071687_

Round 1

Reviewer 1 Report

This review is well written, and represents an overall thorough overview of the topic of inflammation in PD.  Figures are informative and illustrative.  However, some more emphasis could have been placed on certain topics, such as TLRs and non-microglial myeloid cells, as well as the recently discovered link of Parkin/PINK1 deficiency with mtDNA release and resultant systemic inflammation.

In addition:

In Figure 2, the arrow in SCFAs should be pointing up, as SCFAs are directly implicated in enhancing neuroinflammation.

Line 373, it should be PARK6

The role of non-microglial myeloid cells is hardly mentioned, in particular in relation to LRRK2.

There should be more about the role of cells infiltrating the CNS, such as T cells, in experimental PD models, in particular in relation to alpha-synuclein.

Figure 3.  There is nothing in the Figure legend for D.

Line 561, species, not spices

Table 1 is rather uninformative. More details should be provided. It would be better to have preclinical and clinical studies separately.  Results of the studies, when available, should be summarized. Some of the treatments, such as GDNF or alpha-synuclein-targeting strategies, do not belong to this list, as they are not primarily anti-inflammatory.

Author Response

COMMENT: we greatly appreciate th constructive comments of th reviewer. We think that her/his suggestions have improved this manuscript.

This review is well written, and represents an overall thorough overview of the topic of inflammation in PD.  Figures are informative and illustrative.  However, some more emphasis could have been placed on certain topics, such as TLRs and non-microglial myeloid cells, as well as the recently discovered link of Parkin/PINK1 deficiency with mtDNA release and resultant systemic inflammation.

ANSWER: Thanks for this suggestion. We have included a long paragraph at the beginning of chapter 6 to comment the paper by Sliter et al about the role of Parkin and PINK1 in mitigation of STING-induced inflammation.

In addition:

In Figure 2, the arrow in SCFAs should be pointing up, as SCFAs are directly implicated in enhancing neuroinflammation.

ANSWER: We appreciate this valuable comment from the reviewer.  In the study by Sampson and coworkers (ref 116), the authors show restored motor symptoms, microglial activation and reduced α-Syn accumulation when supplementing with SCFAs. On the other hand, they report aggravated Parkinsonian phenotype when transplanting gut microbiota from PD patients. Therefore, one may think that upregulated SCFA levels may enhance inflammation in PD. However, the study by Unger and colleagues showed reduced SCFA levels in a small cohort of PD patients when compared to age-matched individuals (ref 107). Hence, we consider that it may be altered SCFAs composition and not SCFA overall levels which have an impact in PD inflammation. Therefore, we have changed the arrow into ‘altered’ in the figure to avoid overstatements.

 Line 373, it should be PARK6

ANSWER: Corrected.

The role of non-microglial myeloid cells is hardly mentioned, in particular in relation to LRRK2.

ANSWER: Thank you for this useful comment. In Chapter 6, a paragraph and several references have been included regarding the role of LRRK2 in systemic inflammation (ref. 133-136).

There should be more about the role of cells infiltrating the CNS, such as T cells, in experimental PD models, in particular in relation to alpha-synuclein.

ANSWER: Thank you for this useful comment. At the end of Chapter 5, a paragraph and two references (130 and 41) have been included to illustrate the impact of T cell infiltration in the PPF-a-Syn mouse model.

Figure 3.  There is nothing in the Figure legend for D.

ANSWER: Thanks or noting this. The legend for D is now added.

Line 561, species, not spices

ANSWER: corrected

Table 1 is rather uninformative. More details should be provided. It would be better to have preclinical and clinical studies separately.  Results of the studies, when available, should be summarized. Some of the treatments, such as GDNF or alpha-synuclein-targeting strategies, do not belong to this list, as they are not primarily anti-inflammatory.

ANSWER: The studies about alpha-Synuclein and GDNF have been removed from the table and the comment on GDNF has been removed from the main text. The table has been divided into Table 1 for preclinical and Table 2 for clinical studies. We have added a column to summarize the most important results. In the case of the clinical trials, most studies either did not disclose results or are still recruiting. We have removed the trials that, despite being completed, did not publish results. Although the trials at the level of recruiting may seem very preliminary, we think that it is good to include them just to show that this is a very active field.

Reviewer 2 Report

The author, Drs Pajares et al., described a review about the neuroinflammation in Parkinson’s disease. They reviewed the related mechanisms and therapeutic implications.

Minor points

Several references are too old and should be updated. Moreover, the quality of written English needs some language corrections.

Author Response

COMMENT: we greatly appreciate the constructive comments of the reviewer. We think that her/his suggestions have improved this manuscript.

The author, Drs Pajares et al., described a review about the neuroinflammation in Parkinson’s disease. They reviewed the related mechanisms and therapeutic implications.

Minor points: Several references are too old and should be updated. Moreover, the quality of written English needs some language corrections.

ANSWER: we have removed or replaced references published before year 2000. English style has been revised.

Reviewer 3 Report

Pajares and colleagues submitted a well-written review on a relevant topic. The review is comprehensive and summarizes most of the relevant recent literature, but there are some aspects that should be discussed in more details.

Comments:

  • Introduction, first paragraph: the work by Stolzenberg et al. (J Innate Immun 2017) should be briefly summarized when mentioning the role of alpha-Syn in inflammation (even though the authors reference this work also later in their manuscript)
  • Chapter 5, page 7-9: the role of the enteric nervous system and its interaction with the immune system should be discussed more deeply.
  • Chapter 5: The study by Garrido-Gil nicely demonstrated the reciprocal interaction between CNS and gut; the authors might want to discuss this study in more details
  • Chapter 5: Calprotectin has been reported as an inflammatory marker that is elevated in PD (Schwiertz et al.; Mulak et al. …) and should be mentioned, especially because the authors also mention the link between PD and inflammatory bowel disease
  • Chapter 6, page 9: The authors should mention the relevance of LRRK2 in other autoimmune disorders
  • Chapter 8, page 12: “Dopamine can be also produced by neurons from the enteric system [179] and peripheral dopamine levels may be influenced, thus, by gut microbiota.” The authors should briefly discuss whether this leads to relevant systemic concentrations of dopamine.
  • Given the antidiabetic drugs mentioned in Table 1, a paragraph on the link between PD, diabetes mellitus and inflammation would be informative.
  • The authors discuss short chain fatty acids; are there also studies investigating anti-inflammatory effects of this bacterial product in PD (like in other conditions)?

Minor issues:

  • Title: Regarding the state of knowledge concerning inflammation in PD and the authors conclusions, I wonder whether the title should be “Inflammation in …” instead of “Neurinflammation in …”
  • Introduction, first paragraph: There are also other frequent non-motor symptoms in PD and not only neuro-psychiatric symptoms; the authors should briefly mention this.
  • Page 2, line 51: typo; Helicobacter pylori instead of Helicobacter pillory
  • Page 15, line 637: typo; “or by” instead of “of by”

Author Response

COMMENT: we greatly appreciate the constructive comments of the reviewer. We think that her/his suggestions have improved this manuscript.

Pajares and colleagues submitted a well-written review on a relevant topic. The review is comprehensive and summarizes most of the relevant recent literature, but there are some aspects that should be discussed in more details.

Comments:

  • Introduction, first paragraph: the work by Stolzenberg et al. (J Innate Immun 2017) should be briefly summarized when mentioning the role of alpha-Syn in inflammation (even though the authors reference this work also later in their manuscript).

ANSWER: This study is now briefly summarized in Introduction (ref. 8) to state that α-Syn participates in the inflammatory responses of the GI to viral infections and thus provides systemic and brain inflammatory response in PD pathogenesis.

  • Chapter 5, page 7-9: the role of the enteric nervous system and its interaction with the immune system should be discussed more deeply.

ANSWER: Thank you. More introductory background is provided about the gut-brain axis and the immune response. Two references have been included (refs. 99 and 100).

  • Chapter 5: The study by Garrido-Gil nicely demonstrated the reciprocal interaction between CNS and gut; the authors might want to discuss this study in more details

ANSWER: There is now a paragraph describing this study in detail (ref 101).

  • Chapter 5: Calprotectin has been reported as an inflammatory marker that is elevated in PD (Schwiertz et al.; Mulak et al. …) and should be mentioned, especially because the authors also mention the link between PD and inflammatory bowel disease.

ANSWER: Thank you for this comment. Both references have been discussed and included in Chapter 5 (refs. 109 and 110).

  • Chapter 6, page 9: The authors should mention the relevance of LRRK2 in other autoimmune disorders.

ANSWER: This comment has been addressed in the context of the GWAS analysis reported by Witoelar et al, 2017 (ref. 134).

  • Chapter 8, page 12: “Dopamine can be also produced by neurons from the enteric system [179] and peripheral dopamine levels may be influenced, thus, by gut microbiota.” The authors should briefly discuss whether this leads to relevant systemic concentrations of dopamine.

ANSWER: Thank you for this comment. While the role of dopamine in systemic inflammation is widely recognized, only recently it was shown mechanistically that dopamine inhibits TRAF6-mediated NF-κB activation and inflammation via the D5 dopamine receptor in macrophages (ref. 185). Through this mechanism, dopamine protected mice against S. aureus-induced sepsis and meningitis. In another recent study, it was reported that myeloid-specific dopamine D2 receptor signaling controls inflammation in acute pancreatitis via inhibiting M1 macrophage polarization (ref. 186). We have also answered this question in a wider context by considering dysautonomias that may result from alterations of the autonomic nervous system and are responsible at least in part of the non-motor alterations observed in PD (Refs 187, 188 and 189).

  • Given the antidiabetic drugs mentioned in Table 1, a paragraph on the link between PD, diabetes mellitus and inflammation would be informative.

ANSWER: A new paragraph has been added in Chapter 10, to explain the connection between Diabetes and PD and justify the use of PPARgamma agonist in anti-inflammatory therapy of PD (refs. 247 to 253).

  • The authors discuss short chain fatty acids; are there also studies investigating anti-inflammatory effects of this bacterial product in PD (like in other conditions)?

ANSWER: Thanks for this comment. Several reports indicate that SCFAs attenuate the inflammatory response of macrophages in several diseases. In the context of PD, Sampson and coworkers (ref 116) have also reported attenuation of pro-inflammatory response of microglia. However, and aggravated Parkinsonian phenotype is observed when transplanting gut microbiota from PD patients (ref. 107). We have introduced a short discussion about this issue at the end of Chapter 5 and suggest that SCFAs composition, rather than overall SCFA levels, has an impact in PD inflammation

Minor issues:

  • Title: Regarding the state of knowledge concerning inflammation in PD and the authors conclusions, I wonder whether the title should be “Inflammation in …” instead of “Neurinflammation in …”

ANSWER: We have changed “Neuroinflammation” to “Inflammation” in the title and in the running title.

  • Introduction, first paragraph: There are also other frequent non-motor symptoms in PD and not only neuro-psychiatric symptoms; the authors should briefly mention this.

ANSWER: We have included other symptoms such as anosmia and constipation and included a reference about the etiopathogenesis of PD (ref. 2)

  • Page 2, line 51: typo; Helicobacter pylori instead of Helicobacter pillory

ANSWER: corrected

  • Page 15, line 637: typo; “or by” instead of “of by”

ANSWER: corrected